# Adolescent hockey players' predispositions to adopt sport and exercise behaviours: An ecological perspective

Vincent Huard Pelletier, Stephanie Girard, Jean Lemoyne[ID]*

Department of Human Kinetics, Université du Québec à Trois-Rivières, Quebec, Canada

* jean.lemoyne@uqtr.ca

**Data Availability Statement:** All relevant data are within the paper and its Supporting Information files.

## Abstract

Organized sport yields many cognitive, social and physical benefits and is one of the most popular types of physical activity for children and adolescents. Despite the benefits of sports participation, a substantial proportion of adolescents fail to meet Canadian guidelines regarding physical activity. In this regard, it is relevant to understand the mechanisms underlying the adoption of various active behaviours. This study aims to identify the predisposing, enabling and reinforcing factors that potentially influence 4 categories of active behaviours using the Youth Physical Activity Promotion model (YPAP). Data was drawn from 416 male adolescent hockey players ($M_{age}$ = 15.4; $SD$ = 2) who completed a pre-validated questionnaire. Structural equation modeling and interaction analyses were performed to explain the contribution of each determinant. Findings reveal that there are different behavioural patterns based on the type of activity. The interaction between attitudes and environmental factors was a key predictor for each type of behaviour. Perceived competence was associated with more recreational activities, whereas the support of parents and coaches determined involvement in ice hockey. This study refined our understanding of physical activity participation among adolescents already involved in organized sports and emphasized the importance of considering multiple factors surrounding their environment. Several practical recommendations are made to improve young athletes' predisposition to practice physical activity in an organized sports setting.

## Introduction

One of the most popular ways adolescents can meet PA recommendations is through organized sports. Sports are known to enhance physical, socio-affective and cognitive life skills [1], which may explain why approximately 75% of young Canadians take part in them [2]. Although ice hockey is Canada's national sport, participation has sharply dropped by approximately 20% at the end of adolescence [3,4]. Among other factors, organizational barriers and early sports specialization may lead to sports dropout on a long-term basis [5]. On the other hand, diversifying sport practice at a young age has many lasting advantages and may even increase sport success in adulthood [6]. People who take part in organized sports can develop

**Funding:** The author(s) received no specific funding for this work.

**Competing interests:** The authors have declared that no competing interests exist.

abilities and competencies that encourage them to maintain an active lifestyle after their athletic "career" has ended. In Canada, the most popular forms of PA for teenagers are organized sports and fitness activities such as running and weight training [7]. Accordingly, it is relevant to identify the mechanisms that promote teenagers' involvement in various types of active behaviours. In keeping with the "sport sampling" hypothesis [6], it is also relevant to sensitize young hockey players to the possibility of engaging in other types of exercise, and to encourage them to remain more active after their career in organized sport is over.

Previous research shows that active behaviours are determined by multiple factors [8]. In the specific area of physical activity, the Youth Physical Activity Promotion Model [9] offers a useful theoretical framework for studying social, psychological and environmental PA correlates from an ecological perspective. The four correlates in the YPAP (Fig 1) are predisposing, enabling, reinforcing and sociodemographic variables[9]. Predisposing factors are the core element of the YPAP and generally represent the individual and the psychosocial dimensions of the model. More specifically, they are associated with an individual's perceptions and motivation towards PA[9]. According to Welk [9], this construct has two sub-dimensions: "Am I able?" and "Is it worth it?" The "Am I able" component reflects the perceived ability to perform the behaviour. Indeed, the more competent an individual feel in relation to a specific behaviour, the more predisposed he will be to adopt it [9]. The "Is it worth it?" component represents an

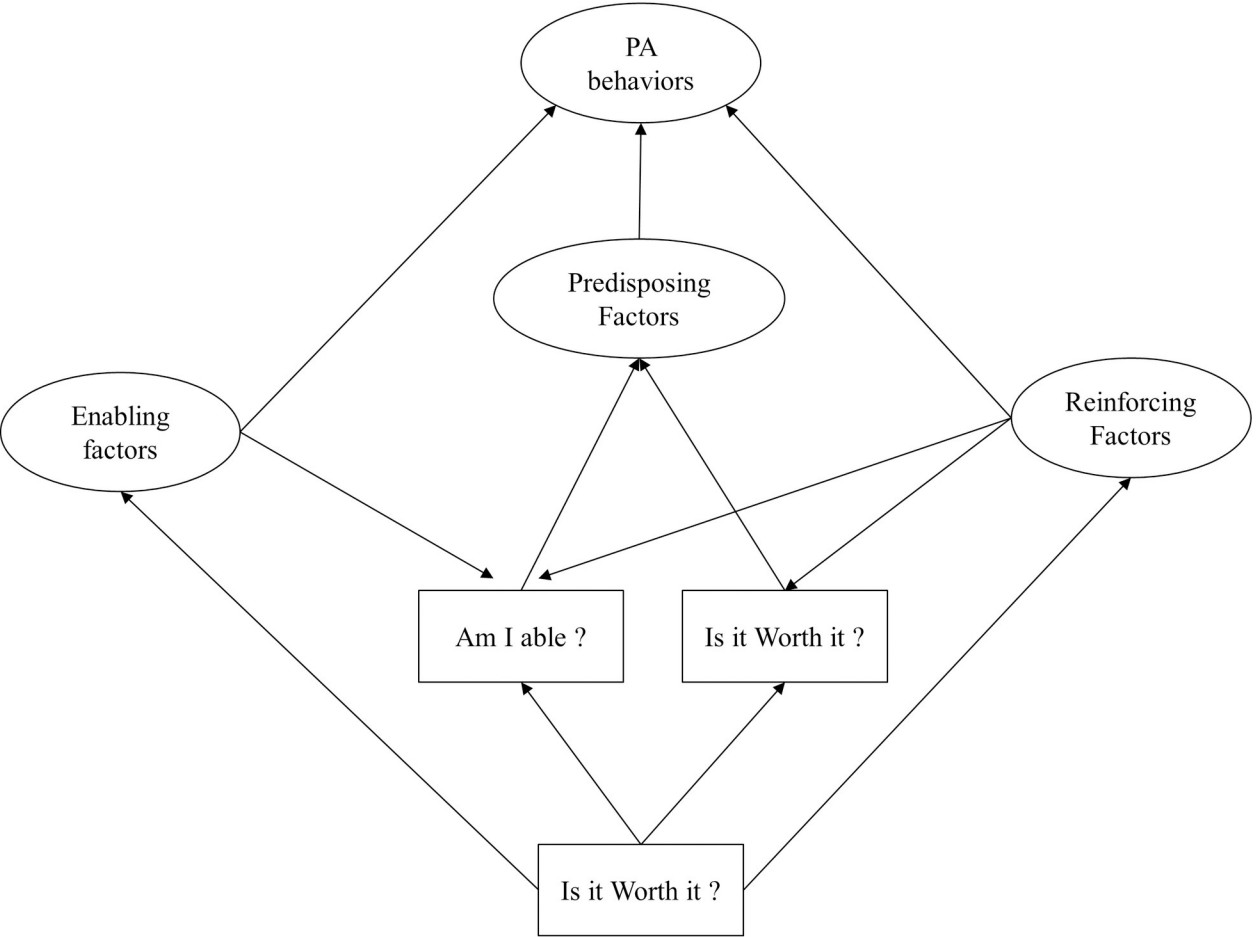

**Fig 1. The Youth Physical Activity Promotion Model (adapted from Welk, 1999).** PA: physical activity.

affective dimension and relates to an individual's evaluation of behaviour. In this sense, an individual's system of beliefs and attitudes regarding behaviour (importance, fun, etc.) will determine his predisposition to adopt it. Enabling factors are environmental variables that may facilitate the adoption of PA practice. Multiple factors can be considered "enabling" and depend on the context in which the behaviour occurs [9]. For example, access to infrastructure, opportunity to practice, skills and fitness were enabling factors used in previous YPAP studies [10]. According to YPAP assumptions, reinforcing factors can potentially influence (favourably or not) the "Am I able" component of the YPAP model [9]. Reinforcing factors are defined as social influences that potentially reinforce predisposing factors [9]. The YPAP model also suggests that reinforcing factors are related to engagement in PA behaviours. For example, parents may be required to transport their children and encourage them constantly to take part in a specific activity (e.g., playing organized hockey) [11]. Sociodemographic variables are also included in the YPAP model, mainly because every other factor (enabling, reinforcing and predisposing) may vary depending on age, sex and socio-economic status [9]. Many authors have supported the relevance of the YPAP over the last few years. Chen et al. demonstrated significant relationships between enabling and predisposing factors [10], and other authors also tested the model as a whole [12]. The ecological perspective of the YPAP model (Fig 1) suggests that each component can interact to influence the adoption of a specific behaviour [9]. However, such interactions have not yet been tested in an organized sports context and with older participants (i.e., 12 years or older).

Considering that many adolescent athletes drop out of organized sport, it is relevant to promote the sampling and experimentation of many types of actives behaviours, which may encourage them to stay physically active once their "athletic career" is over, during their early adulthood. By testing the YPAP model, this study aimed to identify the mechanisms that influence teenage hockey players towards different types of PA behaviours. The first objective was to analyse the contribution of YPAP constructs in four areas of physical activity: 1) organized ice hockey, 2) leisure-time sports, 3) weight training, and 4) cardiovascular training. It is assumed that predisposing (e.g., Am I able, Is it worth it), enabling (environmental influences) and reinforcing (social influences) factors were determinants of PA behaviours. We also hypothesized that enabling and reinforcing factors would influence both predisposing factors (e.g., Am I able, Is it worth it). Because ecological models consider not only the effects of interpersonal and environmental factors, but also the product of interactions between each of these factors [13], the second objective was to test all possible interactions between YPAP determinants.

## Materials and methods

### Research design and participants

We collected data from players in 30 different hockey teams from rural and urban communities in the province of Quebec. Structural equation models with latent variables require large sample sizes to attain a sufficient level of statistical power [14,15]. Therefore, we estimated sample size a priori by calculating the ratio of "cases to free parameters", as suggested by Tanaka [16]. This procedure resulted in a sample size ranging from 275 to 400 participants in order to obtain a satisfactory level of statistical power. The final sample consisted of 416 players (98% males), aged 12 to 17 years old (15.35 ± 1.9 years). We used a convenience sample for the study, but an effort was made to respect the proportions of participants regarding the level of play. The higher number of older players is explained by the fact that players tend to retire at the end of adolescence. We divided participants into three different age groups (Pee-Wee [12–13 years old: n = 94], Bantam [14–15 years old: n = 73], and Midget [16–17 years old: n = 248])

and two levels of play (competitive [n = 213; 51.4%] and recreational [n = 202; 48.6%]). The study was approved by the Comité d'Éthique de le Recherche sur des Êtres Humains de l'UQTR (Approval #: CER-17-240-08-01.10), as well as Quebec's Ice Hockey Federation, which is a provincial branch of Hockey Canada.

We used two strategies to collect data during the 2017–2018 hockey season. For both strategies, all participants and parents of children under 14 gave signed consent forms for their children to participate in the study. For the first data gathering approach, we contacted members of elite hockey programs. All players between 12 and 17 years of age registered in the hockey federation were eligible for the study. We identified teams via the website of Quebec's ice hockey federation, and coaches were phoned or emailed to request permission to meet the players. After explaining the project, research staff members distributed questionnaires to those who agreed to participate, and the participants completed an online or paper version of the questionnaire. The players completed the online version at home while the paper version was completed in a quiet room in the arena before a practice. A second approach involved reaching out to hockey teams directly during tournaments. To proceed, we first asked tournament directors for permission to meet with the teams at the registration desk. The directors gave the researchers permission to meet and inform the coaches, who discussed the project with their players. Research staff members then distributed questionnaires to players who agreed to participate. The questionnaires had to be completed during the weekend in their free time, not immediately before or after a game. They were authorized to complete it in the arena, or in their hotel if the research team was able to pick them up after the team's final game.

## Variables and instruments

A questionnaire was constructed and pre-validated with a pilot sample consisting of 35 local young hockey players between 12 and 17 years of age in the months preceding the study. We performed analyses to assess each scale's psychometric properties (Cronbach alpha and inter item correlations). Preliminary analyses showed acceptable to very good reliability ($\alpha > .70$; inter-item correlation varying between .29 and .70). Tests-retest reliability was not assessed due to the fact that it is difficult to gather a large number of participants during the off-season. The final version of the questionnaire (S1 File) assesses each construct of the YPAP model, resulting in five sub-scales. The following section explains the measurement procedures for these variables.

Sport and exercise behaviours were measured with a physical activity scale that assessed four types of behaviours: 1) organized ice hockey, 2) leisure-time sports, 3) weight training, and 4) cardiovascular activities. Since hockey is played mainly between August and March, we also gathered data about the off-season period. In each category of behaviour, participants were required to indicate their level of participation in terms of frequency (i.e., days or sessions per week). For example, participants were asked: "During a regular season, how many sessions per week (0 to 7) did you spend on weight training?" The same question was repeated for each category of behaviour. This wording allowed us to estimate frequency and type of PA on an annual basis (by assessing for "in" and "off" season). Despite their limitations, self-report measures are the most common way to monitor active behaviours and have demonstrated acceptable reliability [17].

Predisposing factors were divided into two constructs: "Am I able?" (competence) and "Is it worth it?" (attitudes). We assessed the "Am I able" construct using an abbreviated version of the Physical Self-Description Questionnaire [18]. We evaluated three dimensions of physical self-concept reflecting individual perceived abilities towards performing behaviour: perceived

sport competence (4 items), perceived cardiovascular endurance (4 items) and perceived strength (4 items). Each of the sub-scale items was rated on a 6-point Likert scale, (1 = totally disagree; 6 = totally agree). For example, participants were asked to indicate their level of agreement with sentences such as: "Playing sports (or "hockey" in the hockey scale) is for me" (sport competence); "I usually perform well in tasks that require strength" (strength); and, "I can run easily for a long time" (cardiovascular endurance). Preliminary analyses showed very good reliability (McDonald's omega coefficients) for each sub-scale ($\omega_{sport}$ = 0.87 $\omega_{force}$ = 0.85, $\omega_{endurance}$ =.83). For model estimation, each item served as an indicator for its corresponding construct.

The "Is it worth it" construct was measured with a semantic scale of 16 items, which assessed participants' attitudes towards each type of active behaviour (4 items x 4 behaviours). Participants were asked to complete the following sentence: "Playing organized hockey is..." and had to rate their responses on 6-point bipolar adjectives (higher scores equal more attitudes that are favourable): fun-dull, useful-useless, motivating-discouraging, pleasant-unpleasant. Preliminary analyses revealed good reliability for each sub-scale ($\omega_{hockey}$ = .89, $\omega_{weight\ training}$ = .94, $\omega_{cardiovascular}$ = .89 $\omega_{leisure}$ = .88). As with "Am I able", each item served as an indicator for its corresponding construct.

To measure enabling factors, a 9-item, Likert-type scale inspired by Sealens' environmental scale [19] was developed. Preliminary exploratory analyses (principal components analysis) conducted with a pilot sample suggested a two-dimensional structure: 1) access to infrastructure (7 items) and 2) opportunity to take part in PA (2 items). For the 6-point accessibility sub-scale, participants were asked to indicate their level of agreement for items such as: "In my neighbourhood, I have access to an ice rink where I can play hockey" (1 = totally disagree; 6 = totally agree). For the opportunity sub-scale, the same scoring procedure was used, in which participants responded to items such as "At school, they offer me opportunities for weight training." Preliminary analyses showed that the two sub-scales had acceptable reliability ($\omega_{opportunity}$ = .79, $\omega_{accessibility}$ = .69). Two composite scores for accessibility and opportunity were created and used as indicators for the enabling factors construct.

Reinforcing factors were measured by assessing participants' perceptions of parents and coaches towards the practice of one of the four behaviours under study. 2 sub-scales, reflecting parents' and coaches' influences, were created. The "parents" sub-scale consisted of 4 items. The first item asked participants: "How are you motivated to act according to your parents' recommendations?" (1 = totally disagree to 6 = totally agree). The three remaining items measured the behaviours under study: "How often do your parents encourage you to take part in [ice hockey, other sports, weight training and cardiovascular activities]?" The same procedure was repeated for the "coaches" sub-scale. Preliminary analyses showed that the two reinforcing factors sub-scales had good reliability ($\omega_{parents}$ = .76, $\omega_{coaches}$ = .82). Two composite scores (parents and coaches) were used as indicators for the reinforcing factors construct.

## Statistical analyses

Because of the very low level of missing data on most measures (less than 5%), all analyses were conducted with the Expectation Maximization algorithm, which is the default option in Mplus. Normality assumptions were verified with skewness and kurtosis values for each factor. Only attitude toward organized hockey violated normality (skewness = -3.29; kurtosis = 13.79). Mardia's coefficient was also used to evaluate multivariate normality for the overall model, which suggested violation of normality assumptions. Therefore, in keeping with Asparouhov and Muthén [20], the maximum likelihood estimation method was used with the robust estimator (MLR) for the main analyses.

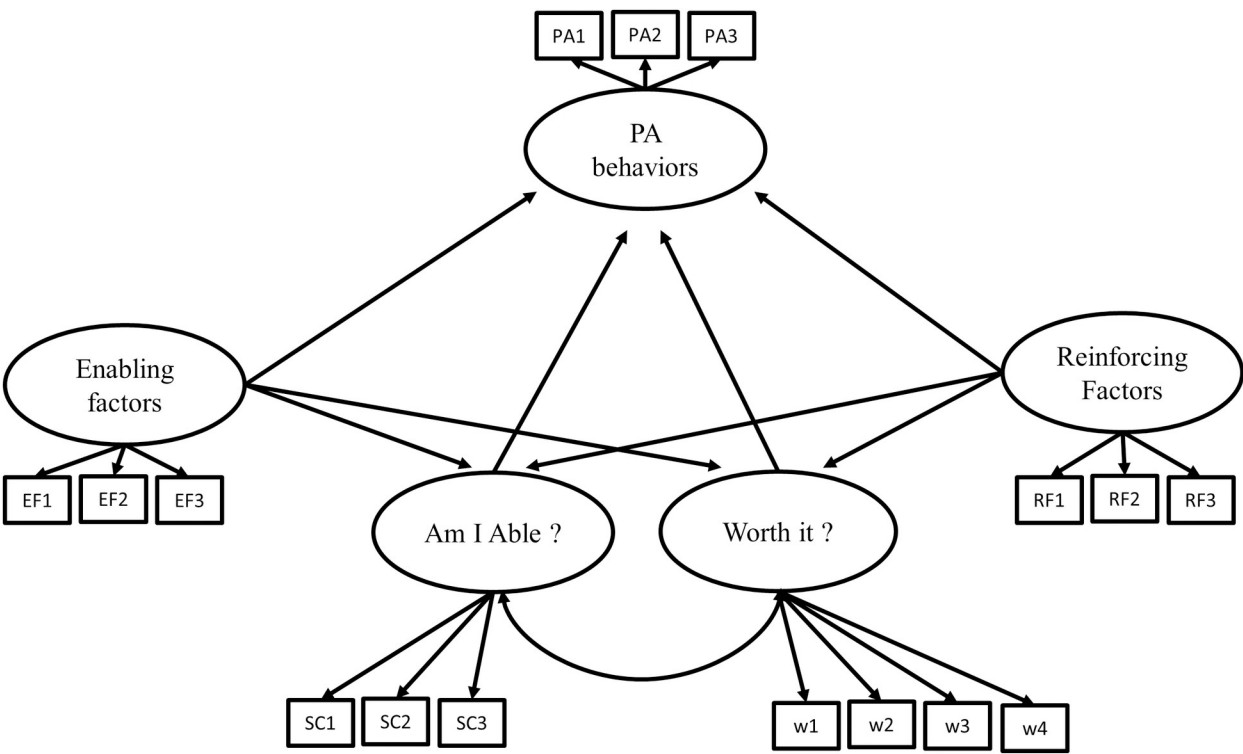

**Fig 2. Model identification.** EF: enabling factors; RF: reinforcing factors; SC: self-concept; w: worth (attitude); PA: physical activity.

Structural equation modeling (SEM) was used to test relationships between YPAP model's constructs. This approach is advantageous in the modelling of relationships involving latent variables and their interaction [20]. To evaluate each determinant's contribution to different active behaviours, models for four categories of active behaviour were tested with SEM: hockey, leisure-time sports, weight training, and cardiovascular activities. Hypothesized relationships were based on the theoretical foundations of the YPAP model (Fig 2). For each model, all YPAP constructs (Am I able, Is it worth it, enabling and reinforcing factors) were latent variables. However, for the "Am I able" and "Is it worth it" constructs, their corresponding behaviour-specific items were used as indicators. The Mplus syntax is presented in S1 Fig. To assess the adequacy of each model, interpretation was based on the cut-off values suggested by Hu and Bentler[15]: 1) chi-square ($\chi 2$) as a global estimation of fit; 2) Comparative Fit Index and Tucker-Lewis Index (CFI and TLI, which should be > .95); and 3) Root Mean Square Error Approximation (RMSEA, which should be < .07). The LaGrange Multiplier (LM) test was also considered, but only for modifications supported by theoretical assumptions. To verify if the new model was significantly better than the previous one, the Satorra-Bentler scaled chi-square difference test was used ($_{SB}\Delta\chi^2$) [21].

## Estimation of interactions between YPAP constructs

The Latent Moderated Structural Equations (LMS) method [22] was used to test interaction effects between latent variables because LMS considers the non-normality of latent variables and their indicators [22]. Because the addition of the numerical integration algorithm (see S2 Fig) prevents calculation of the model fit indices, the log-likelihood ratio test was used to evaluate if the model including the interaction had an improved fit [22]. Detailed tutorials to apply

this method are available [23]. In each model, six interactions were tested one by one: 1) Worth x Enabling, 2) Worth x Reinforcing, 3) Able x Enabling, 4) Able x Reinforcing, 5) Worth x Able, and 6) Enabling x Reinforcing. Next, significant interactions were interpreted using the Johnson–Neyman plots [24,25] (available in Mplus) because they provide information, through regions of significance, on how the relationship between independent and dependent variables changes according to the continuous value of the moderator [26].

## Results and discussion

### Descriptive statistics

Table 1 presents descriptive statistics regarding each of the model's variables. As expected, the most popular in-season activity was ice hockey. Participants' favourite off-season activity was active leisure-time activities, with almost three sessions per week. We verified for age and competitive level differences. Age group comparisons revealed that older players reported higher levels of participation in exercise behaviours ($t_{off}$ = 2.98; $t_{in}$ = 2.83. both at $p < .01$) and showed stronger attitudes (Is it worth it) towards weight training: $t = 3.58$; $p < .001$. Participants who evolved in competitive levels showed a higher level of physical activity than those who were in the recreational level ($t > 2.72$; $p < .01$). Players from competitive levels also showed better attitudes (Is it worth it) on each behaviour: $t > 1.97$; $p < .05$. However, there were not significant differences in regard to their perceived competence ($t < 1.1$; $p > .10$) except for perceived endurance ($t = 2,12$; $p = .034$).

### Model estimation

Table 2 shows the models' fit indices for each type of behaviour. The first three models revealed acceptable-to-good adjustment without the requirement of additional parameters. For model 4, the LM test suggested the addition of one covariance involving two perceived endurance items (end3 and end4). Subsequent to this addition, the model improved significantly ($\Delta_{SB}\chi2$ = 188.97 (1), $p < .001$). Each of the models explained a small-to-moderate proportion of behaviour variance.

The interaction between "Is it worth it" and enabling factors reached statistical significance in each model. This addition improved each model's fit and explained additional variance. Table 3 displays regression coefficients, results of the log-likelihood ratio test and explained variance added by the interaction for all models. According to Martins et al [27], given the decrease in statistical power associated with testing interactions in non-experimental research, the alpha used to infer the statistical significance of interaction was relaxed to $p < .10$. From this perspective, the only case was observed with Model 3 ($p = .056$). According to Kirk [28], the presence of a significant interaction effect suggests it may be biased by the presence of an additional variable. It is therefore recommended to avoid interpreting the "individual" effects

**Table 1. Descriptive statistics according to participants' profile.**

| Constructs-behaviours | Organized Ice HockeyM(SD) | Leisure time sportsM(SD) | Weight trainingM(SD) | Cardiovascular activitiesM(SD) |
|---|---|---|---|---|
| Is it worth it ? | 5.63(.70) | 5.42(.71) | 4.82(1.07) | 4.65(1.07) |
| Am I able ? | 4.89(.88) | 4.89(.88) | 4.12(1.07) | 3.93(.98) |
| Enabling | 4.84(1.22) | 4.73(1.23) | 4.53(1.29) | 4.73(1.11) |
| Reinforcing | 5.01(1.70) | 5.01(1.70) | 4.57(1.25) | 4.57(1.55) |
| Sessions per week | 2.32(1.21) | 2.27(1.35) | 1.86(1.74) | 1.97(1.67) |

M = Mean, SD = Standard deviation.

**Table 2. Fit indices for each model.**

|  | Model 1 Organized Ice Hockey | Model 2 Leisure- time sport | Model 3 Weight training | Model 4 Cardiovascular activity |
|---|---|---|---|---|
| $\chi^2(df)$ | 78.12(56)* | 64.16(56) | 120.39(56)** | 102.41(55)** |
| CFI | .981 | .994 | .972 | .967 |
| TLI | .973 | .991 | .961 | .963 |
| RMSEA | .031 | .019 | .053 | .046 |
| $R^2$(%) | 13 | 7 | 31 | 16 |

*$p < .05$

**$p < .01$ CFI = Comparative Fit Indice, TLI = Tucker Lewis Index, RMSEA = Root Mean Square of Approximation, $R^2$ = Coefficient of determination, $\chi^2$ = Khi$^2$, df = Degree of freedom

of the interacting variables [29]. Among the direct relationships not involved in the interaction effects, "Am I able" was positively associated with participation in leisure-time sports (Model 2) and cardiovascular activities (Model 4). Moreover, reinforcing factors (encouragement by coaches and parents) were a significant predictor for hockey participation (Model 1). Finally, reinforcing factors were positively associated with "Is it worth it" for engagement in weight training (Model 3).

JN plots to interpret significant interactions that were found in each model (Fig 3). To explain participation in hockey (Fig 3A), the adjusted effect of the "Is it worth it" factor was significant only for players reporting enabling factors scores from 0 to 3.1 *SD*s above the mean. For all other participants, the relationship was not significant. In other words, offering opportunities and accessibility to take part in ice hockey enhances the positive relationship between positives attitudes towards the behaviour and participation in hockey. For leisure-time sports

**Table 3. Parameter estimates (standardized solution) for each model tested.**

|  | Model 1 Hockey | Model 2 LTS | Model 3 WT | Model 4 CV |
|---|---|---|---|---|
| Regression coefficients |  |  |  |  |
| W → PA | .24* | .19** | .47*** | .26*** |
| A → PA | .05 | .14** | .05 | .15** |
| EF → PA | .01 | .09 | .27 | .07 |
| RF → PA | .21** | .02 | -.08 | .10 |
| EF → W | .49*** | .59*** | .15** | .15*** |
| RF → A | .09 | .09 | .06 | -.00 |
| EF → A | .32*** | .33** | .66*** | .43*** |
| RF → W | -.01 | -.04 | .19** | .05 |
| Covariance |  |  |  |  |
| W → A | .12** | .33*** | .33*** | .34*** |
| RF ↔ EF | .51*** | .55*** | .56*** | .50*** |
| Interaction |  |  |  |  |
| [EF x W] → PA | .13* | .14** | .25** | .25** |
| $R^2$(%) | 21 | 14 | 51 | 26 |

*Note.* W = Is it worth it?; A: Am I able?; RF: Reinforcing Factors EF: Enabling Factors, PA: Physical Activity, LTS: Leisure-Time Sports WT: Weight Training, CV: Cardiovascular activities.

* $p < .10$

**$p < 0.05$

***$p > 0.01$

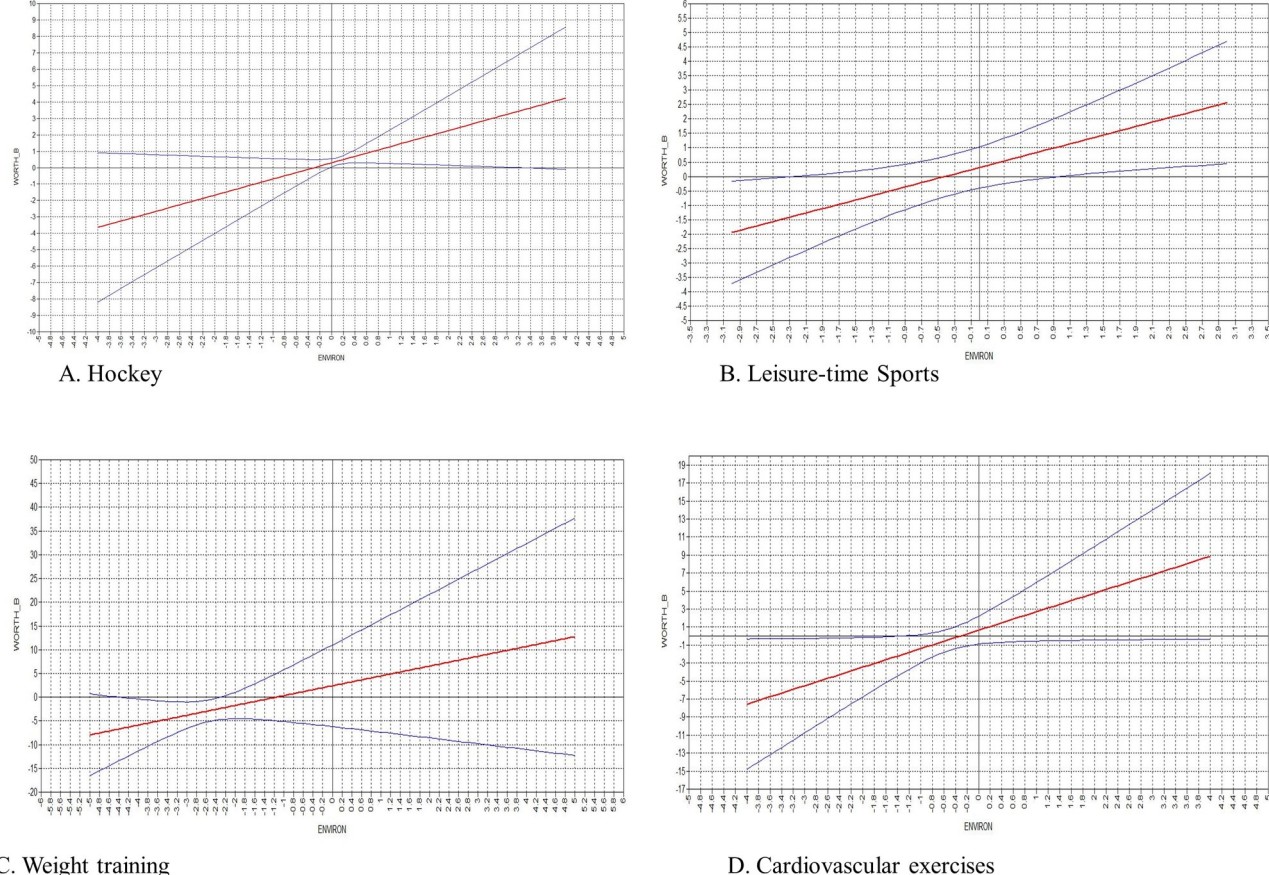

A. Hockey

B. Leisure-time Sports

C. Weight training

D. Cardiovascular exercises

Johnson-Neyman (JN) plots for interpreting interactions involving *Enabling* (x-axis) and *Is it Worth it* factors (y-axis).

**Fig 3. Illustration of Johnson-Neyman plots (for interpreting worth* enabling factors interactions).** 3A. Hockey; 3B. Leisure-time sport; 3C. Weight training; 3D. Cardiovascular activities.

(Fig 3B), the adjusted effect of the "Is it worth it" factor was significant only for participants who scored 2.25 *SD*s below or 1 *SD*s above the mean of enabling factors. This shows that in a favourable environment, attitudes are positively associated with leisure-time sports participation. Moreover, in the absence of a lack of opportunities and accessibility (2.25 *SD*s below the mean), attitudes are important determinants of sports participation. For weight training (Fig 3C), the adjusted effect of the "It is worth it" factor was significant only when the scores of enabling factors scores were between 2.3 *SD*s below the mean. This shows that in the absence of opportunities and accessibility, attitudes are important determinants of weight training. For cardiovascular activities (Fig 3D), the adjusted effect of the "Is it worth it" factor was significant only for participants scoring 1.4 *SD*s below the mean of enabling factors. In other words, in the absence of opportunities and accessibility, attitudes are important determinants of running.

In view of the harmful effects of organized sports dropout, promoting an active lifestyle during adolescence is crucial in terms of education and public health. The purpose of this investigation was to refine our understanding of the mechanisms predisposing young Canadian male hockey players towards participation in four areas of active behaviours. The YPAP model seems like a promising theoretical framework because it allows us to evaluate how

ecological factors (e.g., social influences and built environment) contribute to teenagers' pre-dispositions towards active behaviours. This study further develops the YPAP model because it was conducted among an older population (12 years and older), whereas previous studies mainly focused on younger participants (8 to 12 years old) [10, 30,31]. Next, the YPAP model served to examine specific categories of behaviours, thereby expanding on the knowledge gained from prior studies concerned with global measures of physical activity. Consideration of different types of behaviour, moreover, is especially relevant because it provides more accurate estimates of their determinants. Indeed, it was possible to hypothesize that each determinant had a different role to play regarding the types of behaviour under study.

In the first model (hockey), our findings demonstrate that reinforcing factors play an important role in hockey participation, as previous studies have shown [32,33]. Indeed, they emphasize the importance of considering the support of coaches and parents when motivating adolescents towards their sport. For example, the kind of support young players are given (e.g., encouragement, transportation, positive "player-coach" relationships, etc.) can influence (favourably or not) their willingness to participate in hockey. As for the positive relationship with attitudes (Is it worth it), this varies based on enabling factors (opportunities and access), suggesting that when the environment is favourable, adolescents tend to perceive hockey more positively and engage, as a result, in their sport of choice. These findings agree with those of Yan et al.[34], who showed that access to sport infrastructure facilitated PA adherence. Surprisingly, perceived competence (Am I able) was not significantly associated with participation in ice hockey, which contradicts the findings of antecedent research [10]. It could be explained, however, by the use of a "global" perceived sport competence scale instead of a more specific "perceived hockey competence" scale. Prior research on gymnastics, for example, indicate that a sport-specific scale can help better explain the relationship between self-competence and behaviour [35]. Finally, enabling factors were also associated with players' perceived competence to participate in ice hockey, which underscores the importance of providing opportunities to practice and play hockey.

Interestingly, the second and fourth models (leisure-time sports and cardiovascular activities) displayed similar behavioural patterns. First, the "Am I able" factor was positively associated with both behaviours; this demonstrates the important role played by perceived competence and is consistent with previous studies [10,12]. Relationships with reinforcing factors were not significant, however. The reason could be that the support of parents and coaches, the only reinforcing factor measured in this study, may not be the most influential way to promote multi-sport participation. In fact, other components of social influences reflecting parents' and coaches' support for active behaviours (e.g., transportation to infrastructures, training together, etc.) may need to be considered to effectively assess the contribution of reinforcing factors. A further explanation could be the context in which such behaviours occur. It is plausible that participants in our study took part in leisure-time sports and cardiovascular activities in a recreational context. In this case, a sense of competence in different areas of physical activity often prompts individuals to invest in several types of physical activity (e.g., multiple sports, jogging, etc.) instead of a single one (e.g., hockey only). Second, as the ice hockey model shows, the relationship with attitudes (Is it worth it) and behaviour varied according to the enabling factors. For leisure-time sports practice, attitudes were positively associated with sports participation when the environment was favourable. In contrast, when opportunities and access to sport infrastructure were lacking, attitudes became even more important in predisposing adolescents to engage in sport. For cardiovascular activities, the positive relationship with participants' attitudes (Is it worth it) was significant only when the environment was reported to be unfavourable. In other words, attitudes are a more important determinant of this active behaviour in unsupportive environments.

The third model (weight training) displayed similar results to the other three models. In this sense, the interaction between "Is it worth it" and the environment in which adolescents played was a key determinant of behaviour. The interaction suggests that when access and opportunities are lacking, valuing weight training positively contributes to greater involvement in this form of behaviour. Similar to the hockey model, a sense of competence (Am I able) towards weight training did not guarantee participation in this activity. In this regard, McReary and Sasse [36] showed that young boys' participation in weight training was associated with lower levels of self-esteem. Logically, therefore, their perceived competence has a negligible impact in this area. Moreover, the role of enabling factors was also important in terms of the participants' predisposing factors where it was associated with more positive attitudes and stronger perceived ability. Such conclusions emphasize the importance of offering adolescents different options that could influence them towards weight training. Training sessions at school or opportunities to try out new activities (e.g., Crossfit, Olympic weightlifting, powerlifting), for example, could enhance adolescents' predisposition towards weight training. Differences regarding behavioural patterns between weight training and the previous models may be explained by the context in which weight training takes place for male adolescents. According to Kilpatrick [37], the reasons for adolescent boys' adherence to weight training could involve the desire to enhance their body image or simply improve sports performance. Such questions, however, are beyond the scope of this study. Further analyses should be conducted to verify if the patterns displayed by more competitive hockey players differ from those of recreational hockey players.

## Practical implications

In general, our results showed that a positive attitude towards active behaviours combined with a favourable environment are key elements in the inclination to practice more diverse physical activity. This is in line with previous studies conducted among adolescent sport populations in specific contexts where fostering positive attitudes across a diversity of physical activities is important [38]. Practically speaking, stakeholders (e.g., coaches, parents, teachers, decision makers) should create opportunities leading to positive and pleasant experiences in different areas of physical activity. The importance of perceived competence is also an important aspect, especially because of its association with leisure-time sports and cardiovascular activities. In this regard, fostering a strong self-concept in different behaviours should lead to higher participation. According to Hynynen et al. [39], one way to improve self-concept is to offer participants opportunities to enhance their perceptions of social acceptance, physical appearance and athletic ability. An improved self-concept could even contribute to a higher level of performance in the opinion of Marsh and Perry [40].

Generally speaking, and based on Canadian guidelines [41], active games and playgrounds that are accessible to all should be developed and promoted. Moreover, making young participants aware of existing programs that promote and offer access to a variety of physical activities should become a priority. Hockey federations should be encouraged to plan seasons in a way that allows young Canadians to practice other sports. The Quebec Ice Hockey Federation, for example, recently developed an initiative encouraging teenagers to take part in other types of physical activities [42]. Finally, decision makers should design policies for positive and developmental sport experiences that help prevent or, at least, reduce sport attrition. As might be expected, a variety of strategies can be explored to improve the social environment of young hockey players. To start, young people's long-term involvement in exercise behaviours may increase if they are motivated to participate in multiple forms of active behaviours in their school environment [43]. More certified coaches should be hired and asked to promote

physical literacy in their "in-season" time with the players. Finally, minor hockey associations could try to educate and/or sensitize parents and coaches concerning their potential impact on the players' motivation to participate in different types of active behaviours [44].

## Conclusions

This investigation points to some interesting avenues for future research. The first is to verify whether YPAP model components and their interactions vary when socio-demographic variables are taken into account. If so, more variables should be considered in future research. For example, early sport specialization is frequently reported in ice hockey[5], and it would be advantageous to know if behavioural patterns differ in terms of "early specializers" versus "samplers." Moesch and al.[6] stipulate that early specialization can be detrimental to maintaining a good level of intrinsic motivation and lead to a higher risk of dropout. Other group comparisons regarding participants' age and competition level (at which they are playing) would be useful and offer a more accurate explanation of the impact of social environments on young hockey players. An interesting statistical approach could involve conducting invariance testing on the YPAP model based on categories of variables, which may help improve understanding of the mechanisms underlying adolescents' adoption of multiple active behaviours. Longitudinal designs could also provide further information about the contribution of additional variables to the model during a longer period of time (e.g., during adolescence). From this perspective, it could be beneficial to evaluate how PA determinants evolve over time, thereby gaining a more accurate understanding of how participants transfer from one sport behaviour to another after participation in organized sports has ended. Finally, it would be interesting to examine the relationships between fluctuations in reinforcing factors and specific influences from parents, coaches and peers over time. Such relationships has already been tested in some studies [45–47]. Despite promising results, it has not been tested in an ecological model designed, and was not designed to verify the possible contribution of other variables. Another way to improve our understanding of PA diversity in teenage athletes would involve studying the point of view of parents and coaches and verify how it interact with the environment in which adolescent evolve. This could be achieved through qualitative or mixed designs, where group interviews with parents/coaches would be conducted to investigate their beliefs regarding the diversity of their childrens'/students' PA practice.

### Limitations of the study

The limitations of this study must be highlighted despite its various contributions to the YPAP literature. To begin, PA was assessed with self-reported measures. Self-reported assessments of PA are, regardless of their limitations, the most common way to monitor active behaviours and are effective for measuring adolescents' PA. Although objective measures can provide a more accurate estimate of behaviours, they do not offer detailed information about the variety of PA practiced. However, PA can be monitored in more specific ways in the future; for example, participants could be encouraged to keep a PA log (or diary) on a weekly basis which may minimize the risk of bias[43]. Another limitation concerns the measurement of the "Am I able" factor. In the present investigation, environmental variables were used as a proxy for enabling factors. As in previous studies, the use of some additional indicators for enabling factors (e.g., fitness, ability level) may have provided a more accurate estimate of this determinant, in the manner of Chen et al.[10] These choices were made for reasons of feasibility to create a parsimonious questionnaire. Future research initiatives should consider integrating such additional indicators. Another limitation is sample size. A larger sample size (800–1000 participants) could have significantly reduced the risk of fit indices estimation error[42] and would ascertain

the generalizability of our findings. Finally, although no systematic sampling strategies were used for this study, the distribution of age groups was considered. Given that sport dropout tends to occur toward the end of adolescence, categories corresponding to this age were emphasized. The low proportion of female players was expected and can be explained. The 2% participation rate of females in our study is similar with data from the hockey federation regarding these age categories [4]. Subsequent studies should use systematic sampling procedures to respect proportion based on level of play (e.g., recreational, competitive), gender and other factors associated with hockey participation.

This study focused mainly on examining the different factors that impact the active behaviours of young people. Multiple active behaviours were successfully explained with the help of YPAP determinants. In summary, positive attitudes towards active behaviours combined with the presence of opportunities and access to infrastructures are crucial for promoting a diversity of active behaviours. This study refined our understanding of the YPAP and its possible applications among adolescent hockey players. Although further studies are needed to fully explain young athletes' engagement in a variety of physical activities, our study creates a solid foundation for future researchers.

## Supporting information

**S1 File. YPAP questionnaire.**
(TIF)

**S1 Fig. Syntax for model estimation.**
(TIF)

**S2 Fig. Syntax for interaction testing.**
(TIF)

## Acknowledgments

The authors are grateful to Hockey Quebec for supporting the project. We also thank all participating teams/players who made this research possible and all players who took the time to complete the questionnaires.

## Author Contributions

**Conceptualization:** Vincent Huard Pelletier.

**Methodology:** Vincent Huard Pelletier, Stephanie Girard.

**Software:** Stephanie Girard, Jean Lemoyne.

**Supervision:** Jean Lemoyne.

**Validation:** Vincent Huard Pelletier.

**Writing – original draft:** Vincent Huard Pelletier.

**Writing – review & editing:** Vincent Huard Pelletier, Stephanie Girard, Jean Lemoyne.

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
