## [Decision Letter · Decision Letter 0]

15 Oct 2019

PONE-D-19-23863

DIVERSIFICATION OF YOUNG HOCKEY PLAYERS’ ACTIVE BEHAVIORS: CONTRIBUTION OF INDIVIDUAL, SOCIAL, AND ENVIRONMENTAL FACTORS

PLOS ONE

Dear Prof. Lemoyne,

Thank you for submitting your manuscript to PLOS ONE. After careful consideration, we feel that it has merit but does not fully meet PLOS ONE’s publication criteria as it currently stands. Therefore, we invite you to submit a revised version of the manuscript that addresses the points raised during the review process.

We would appreciate receiving your revised manuscript by Nov 29 2019 11:59PM. To enhance the reproducibility of your results, we recommend that if applicable you deposit your laboratory protocols in protocols.io, where a protocol can be assigned its own identifier (DOI) such that it can be cited independently in the future. For instructions see: http://journals.plos.org/plosone/s/submission-guidelines#loc-laboratory-protocols

We look forward to receiving your revised manuscript.

Kind regards,

Francisco Javier Huertas-Delgado, Ph.D.

Academic Editor

PLOS ONE

Journal Requirements:

2. In your Methods section, please provide additional information about the participant recruitment method and the demographic details of your participants. Please ensure you have provided sufficient details to replicate the analyses such as: a) the recruitment date range (month and year), b) a description of any inclusion/exclusion criteria that were applied to participant recruitment, c) a table of relevant demographic details, d) a statement as to whether your sample can be considered representative of a larger population, e) a description of how participants were recruited, and f) descriptions of where participants were recruited and where the research took place.

3. Please include additional information regarding the survey or questionnaire used in the study and ensure that you have provided sufficient details that others could replicate the analyses. For instance, if you developed a questionnaire as part of this study and it is not under a copyright more restrictive than CC-BY, please include a copy, in both the original language and English, as Supporting Information. Moreover, please include more details on how the questionnaire was pre-tested, and whether it was validated.

4. We note that Figure 1 in your submission contain copyrighted images. All PLOS content is published under the Creative Commons Attribution License (CC BY 4.0), which means that the manuscript, images, and Supporting Information files will be freely available online, and any third party is permitted to access, download, copy, distribute, and use these materials in any way, even commercially, with proper attribution. For more information, see our copyright guidelines: http://journals.plos.org/plosone/s/licenses-and-copyright.

5. Please amend the manuscript submission data (via Edit Submission) to include author MSc, Stephanie Girard

Additional Editor Comments (if provided):

Reviewers' comments:

Reviewer's Responses to Questions

**Comments to the Author**

1. Is the manuscript technically sound, and do the data support the conclusions?

Reviewer #1: Partly

Reviewer #2: Yes

Reviewer #3: Yes

2. Has the statistical analysis been performed appropriately and rigorously? 

Reviewer #1: No

Reviewer #2: Yes

Reviewer #3: Yes

3. Have the authors made all data underlying the findings in their manuscript fully available?

Reviewer #1: Yes

Reviewer #2: Yes

Reviewer #3: Yes

4. Is the manuscript presented in an intelligible fashion and written in standard English?

Reviewer #1: Yes

Reviewer #2: Yes

Reviewer #3: Yes

5. Review Comments to the Author

Reviewer #1: The present study aimed to investigate: “DIVERSIFICATION OF YOUNG HOCKEY PLAYERS’ ACTIVE BEHAVIOURS: 4 CONTRIBUTION OF INDIVIDUAL, SOCIAL, AND ENVIRONMENTAL FACTORS”

Follow bellow my considerations:

1) Abstract: adjust numbers after comma. Insert the main results in the abstract.

2) Introduction: several phrases don’t have references. Please, add it.

3) Methods: Is the authors compared the responses of age rank, competitive level, and sexes? Are the statistical analyses employed was enough to support the results?

4) Discussion: please, change in all information in the article the term “we”. Change to it was or other.

5) What the reproducibility of the fill these questionnaires? Are the authors measured it?

6) The main point refers to aggregate all adolescents in a unique group. I believe that the groups should be split in competitive level and sex.

7) Is the sample size for woman enough? Is represented the practitioners in Canada?

Reviewer #2: This manuscript used a cross-sectional descriptive study to investigate the predisposing, enabling, and reinforcing factors that potentially influence active behaviours in adolescents.

The topic of the manuscript is appropriate for the Journal.

I believe the study findings are of great interest to investigators and stakeholders.

This paper would benefit from some minor English language editing, as few grammatical errors were noted thought the manuscript.

Materials and Methods: The main comment I leave is related to the procedure for applying the questionnaire. I ask the authors to consider a possible bias in the results based on the questionnaires collected in the second approach (during the tournaments). Having been the questionnaires applied at the end of the final game, could not have influenced some of the answers? Can this be considered a limitation of the study?

Sufficient details about the process are provided. Statistical analyses used are appropriate. The methods are appropriate and well described.

Results: Information is clearly provided on tables; however, the pictures appeared in the document with poor quality, hindering a more accurate visualization of the results.

Figures and tables: The three tables are clear and well designed. The three figures are with low definition impairing reading, especially figure 3.

References: There are 42, and all seem appropriate.

Reviewer #3: The present paper is interesting and well-structured. However, there are few issues that are suggested to be addressed to the manuscript prior to publication:

It is suggested to rewrite the title, is big and not attractive.

The abstract at some point does not make sense. First the authors expose that there is a substantial proportion of adolescents that don’t meet Canadian guidelines to have a good levels of physical activity, and then the authors want to know more about the physical activity participation of adolescents already involved in an organized sport?

Why authors did not consider add girls to the study’s sample?

6. PLOS authors have the option to publish the peer review history of their article (what does this mean?). If published, this will include your full peer review and any attached files.

Reviewer #1: Yes: Braulio Henrique Magnani Branco - Ph.D. Professor at University Center of Maringa - Post-graduation Program of Health Promotion.

Reviewer #2: No

Reviewer #3: No

---

## [Author Response · Author response to Decision Letter 0]

22 Nov 2019

We provided a completed an "authors' response" file in the revision. Each comment is adresse, and we provide a clear response for each point.

---

## [Decision Letter · Decision Letter 1]

2 Jan 2020

PONE-D-19-23863R1

ADOLESCENT HOCKEY PLAYERS’ PREDISPOSITIONS TO ADOPT SPORT AND EXERCISE BEHAVIOURS: AN ECOLOGICAL PERSPECTIVE

PLOS ONE

Dear Prof. Lemoyne,

Thank you for submitting your manuscript to PLOS ONE. After careful consideration, we feel that it has merit but does not fully meet PLOS ONE’s publication criteria as it currently stands. Therefore, we invite you to submit a revised version of the manuscript that addresses the points raised during the review process.

We would appreciate receiving your revised manuscript by Feb 16 2020 11:59PM. To enhance the reproducibility of your results, we recommend that if applicable you deposit your laboratory protocols in protocols.io, where a protocol can be assigned its own identifier (DOI) such that it can be cited independently in the future. For instructions see: http://journals.plos.org/plosone/s/submission-guidelines#loc-laboratory-protocols

We look forward to receiving your revised manuscript.

Kind regards,

Francisco Javier Huertas-Delgado, Ph.D.

Academic Editor

PLOS ONE

Reviewers' comments:

Reviewer's Responses to Questions

**Comments to the Author**

1. If the authors have adequately addressed your comments raised in a previous round of review and you feel that this manuscript is now acceptable for publication, you may indicate that here to bypass the “Comments to the Author” section, enter your conflict of interest statement in the “Confidential to Editor” section, and submit your "Accept" recommendation.

Reviewer #2: All comments have been addressed

Reviewer #3: (No Response)

2. Is the manuscript technically sound, and do the data support the conclusions?

Reviewer #2: Yes

Reviewer #3: Yes

3. Has the statistical analysis been performed appropriately and rigorously? 

Reviewer #2: Yes

Reviewer #3: Yes

4. Have the authors made all data underlying the findings in their manuscript fully available?

Reviewer #2: Yes

Reviewer #3: Yes

5. Is the manuscript presented in an intelligible fashion and written in standard English?

Reviewer #2: Yes

Reviewer #3: Yes

6. Review Comments to the Author

Reviewer #2: The authors present improvements in the article that allow the reader to be clarified, especially regarding some methodological aspects.

The authors should review the captions in tables 1 and 2, as they do not allow identifying some abbreviations. More specifically, in table 1, the abbreviations for "M" and "SD" are not defined in the caption. In table 2, are not identified in the caption, the abbreviations corresponding to the variables, which would facilitate the interpretation of the reader.

Reviewer #3: The present paper is interesting and has a significant content. Moreover, it is appeared to be suitable to highlight the premise of this manuscript is a worthy one, and the authors spent a great time in the research and structuring. However, there are some questions that need to be addressed to the manuscript prior to publication.

Introduction

In this section, it is considered that the pertinence of the study needs to be clearer. What is the real importance of the present study?

Methods

The authors should homogenize the terms that use all over the manuscript. In the introduction for the areas of physical activity used the following order and nomenclature: 1) organized ice hockey, 2) leisure-time sports, 3) weight training, and 4) cardiovascular training, however, in the variables and instruments, the authors change the order before and use different terms (e.g.: aerobic capacity). Please uniformize it.

Results

Table 1 need to be checked, there are wrong values (SD = 88 ?, or 1.700?), there also missing parenthesis.

Discussion

The authors mentioned in this section that previous studies mainly focused on younger participantes (8-12 years old), but did not referenced. It is suggested to add those studies mentioned before.

Perspective of future researches

The authors exposed that particular relationship has already been tested in some studies, but they only mentioned one study (Amorose et al., 2016). Please add more studies to justify the sentence.

7. PLOS authors have the option to publish the peer review history of their article (what does this mean?). If published, this will include your full peer review and any attached files.

Reviewer #2: No

Reviewer #3: No

---

## [Author Response · Author response to Decision Letter 1]

10 Jan 2020

Editor will find an updated version of the manuscrit. Every reviewers' comments are annontated and provide response for each point.

---

## [Editor Report · Decision Letter 2]

14 Jan 2020

ADOLESCENT HOCKEY PLAYERS’ PREDISPOSITIONS TO ADOPT  SPORT AND EXERCISE BEHAVIOURS: AN ECOLOGICAL PERSPECTIVE

PONE-D-19-23863R2

Dear Dr. Lemoyne,

We are pleased to inform you that your manuscript has been judged scientifically suitable for publication and will be formally accepted for publication once it complies with all outstanding technical requirements.

With kind regards,

Francisco Javier Huertas-Delgado, Ph.D.

Academic Editor

PLOS ONE

---

## [Editor Report · Acceptance letter]

30 Jan 2020

PONE-D-19-23863R2 

Adolescent hockey players’ predispositions to adopt sport and exercise behaviours: an ecological perspective 

Dear Dr. Lemoyne:

I am pleased to inform you that your manuscript has been deemed suitable for publication in PLOS ONE. Congratulations! Your manuscript is now with our production department. 

With kind regards,

on behalf of

Dr. Francisco Javier Huertas-Delgado 

Academic Editor

PLOS ONE